# Empirical Comparison of Supervised Learning Methods for Assessing the Stability of Slopes Adjacent to Military Operation Roads

SeMyung Kwon [1,†] , Leilei Pan [2,†] , Yongrae Kim [3] , Sang In Lee [3] , Hyeongkeun Kweon [4] , Kyeongcheol Lee [4] , Kyujin Yeom [5] and Jung Il Seo [2,3,*]

1 Division of Administration, Forest Restoration Center, Korea Association of Forest Enviro-Conservation Technology, 150 Osongsaengmyeong 3-ro, Osong-eup, Heungdeok-gu, Cheongju-si 28165, Chungcheongbuk-do, Republic of Korea; ksmeong@gmail.com

2 Department of Forest Science, Kongju National University, 54 Daehak-ro, Yesan-eup, Yesan-gun 32439, Chungcheongnam-do, Republic of Korea; panleisimo@gmail.com

3 Institute of Ecological Restoration, Kongju National University, 54 Daehak-ro, Yesan-eup, Yesan-gun 32439, Chungcheongnam-do, Republic of Korea; canclub2001@gmail.com (Y.K.); sanginlee@kongju.ac.kr (S.I.L.)

4 Department of Crops and Forestry, Korea National University of Agriculture and Fisheries, 1515 Kongjwipatjwi-ro, Deokjin-gu, Jeonju-si 54874, Jeollabuk-do, Republic of Korea; hkkweon00@gmail.com (H.K.); dlrud112@korea.kr (K.L.)

5 CCZ Forest Land Management Office, Korea Forest Conservation Association, 51 Munjeong-ro 40beon-gil, Seo-gu, Daejeon-si 35262, Republic of Korea; nalse@kfca.re.kr

* Correspondence: jungil.seo@kongju.ac.kr; Tel.: +82-41-330-1302

† These authors contributed equally to this work.

**Abstract:** The Civilian Access Control Zone (CACZ), south of the Demilitarized Zone (DMZ) separating North and South Korea, has functioned as a unique bio-reserve owing to restrictions on human use. However, it is now increasingly threatened by damaged land and slope failures. In this study, a machine-learning-based method was used to assess slope stability by introducing the random forest (RF), support vector machine (SVM), extreme gradient boosting (XGBoost), and logistic regression (LR) approaches. These classification models were trained and evaluated on 393 slope stability cases from 2009 to 2019 to assess slope stability in the northern area of the Civilian Control Line, South Korea. For comparison, the performance of these classification models was measured by considering the accuracy, Cohen's kappa, F1-score, recall rate, precision, and area under the ROC curve (AUC). Furthermore, 14 influencing factors (slope, vegetation, structure conditions, etc.) were considered to explore feature importance. The evaluation and comparison of the results showed that the performance of all classifier models was satisfactory for assessing the stability of the slope, the ability of LR was validated (accuracy = 0.847; AUC = 0.838), and XGBoost proved to be the most efficient method for predicting slope stability (accuracy = 0.903; AUC = 0.900). Among the 14 influencing factors, the external condition was the most important. The proposed supervised learning method offers a promising method for assessing slope status, may be beneficial for government agencies in early-stage risk mitigation, and provides a database for efficient restoration management.

**Keywords:** machine learning; slope stability; variable importance; forest restoration management; DMZ

## 1. Introduction

The Republic of Korea is the only divided country in the world; the Republic of Korea and the Democratic People's Republic of Korea have been in a state of conflict for approximately 70 years since the armistice agreement of the 6.25 War (Korean War).

The military demarcation line (MDL) between the two countries is oriented in the east–west direction, with a total length of approximately 238 km. The parallel lines separated by

2 km to the south and north of the Military Demarcation Line are respectively called the Southern Limit Line (SLL) and the Northern Limit Line (NLL). The Demilitarized Zone (DMZ) between these two lines is a place of high historical and ecotourism value as traces of the Korean War and the ecosystem are well preserved. Approximately 5–20 km away from the SLL, there is a civilian control line (CCL). This line forms a buffer zone to the DMZ, namely, the Civilian Access Control Zone (CACZ) (Figure 1). The entrance of civilians into this area is limited (although not as extensively as with the DMZ) to protect and maintain the security of military installations and operations near the DMZ, but limited agricultural activities that do not affect military operations and security are allowed. However, many places have a high probability of landslides due to damage caused for military purposes.

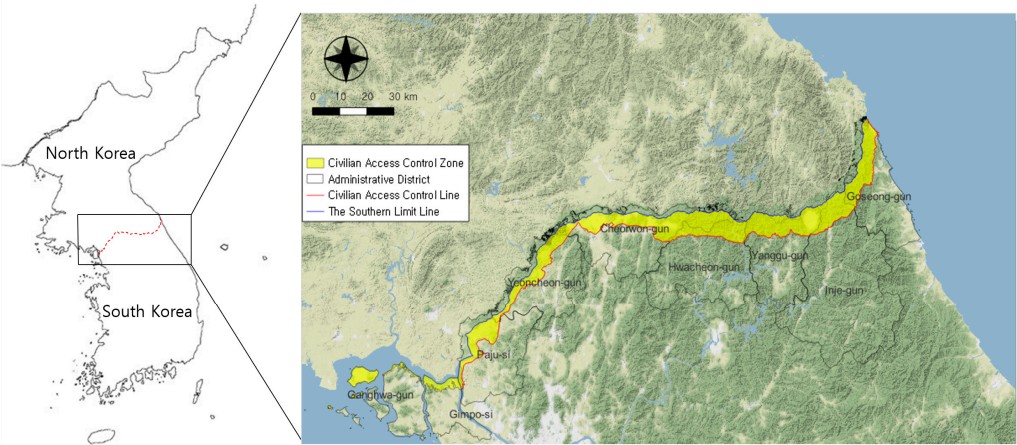

**Figure 1.** The Civilian Access Control Zone (CACZ).

According to the results of the "Mountain Management Survey and Monitoring in the CACZ" conducted by the Korean Forest Service (South Korea), 3681.3 hectares of damaged mountain land were found in the CACZ. For the damage sources, the construction of military installations accounted for the most at 1278.3 ha (34.7%), followed by cultivated land at 1065.1 ha (28.9%) and roads at 599.9 ha (16.3%). Among these, the highest proportion of damage to mountainous areas was caused by the construction of military tactical roads, where forest soil sediment disasters have been reported to occur. Unlike constructing forest roads, when developing tactical roads in the remote and isolated areas of the CACZ, it is difficult to introduce facility materials and vegetation that promote the physical stability of slopes, and the input and utilization of technical personnel are limited.

These limitations have also been confirmed in related studies on the CACZ boreal forest ecosystem. In order to propose mid- to long-term management plans for the protection of native plants from invasive species in the ecosystem, a number of studies have been conducted through vegetation surveys in areas with less human disturbance to identify rare plants, endemic plants, naturalized plants [1–7], and northern lineage plants [8]. The categorization and grading of wetlands formed by topographical characteristics have also been studied by evaluating factors such as vegetation, hydrology, hydraulics, human landscapes, and disturbances [9,10]. However, owing to limited access, some related research on mountain disasters has been carried out by reading satellite images, dividing the damaged area into bare soil, landslide area, poor growth areas, etc. and calculating their areas, thus providing basic information for recovery planning [11,12].

Recently, the Korean Forest Service attempted to develop a practical and effective restoration technique by surveying areas where forest restoration projects had previously been conducted in the CACZ. Through logistic regression (LR) analysis, they concluded that the most important factors to be considered in the process of restoration are the external condition and vegetation coverage [13]. This analysis was limited in that it only used the logistic regression analysis method.

The linear regression (LR) method is the most widely used method for slope stability prediction [14–17] and landslide susceptibility [18–20] in previous studies; however,

as a traditional form of linear regression, LR has certain limitations in determining the important factors affecting the stability of the restoration area. It is difficult for a single learning algorithm to cope with complicated learning problems with high performance. Furthermore, the classical statistical regression method does not effectively describe the complex nonlinear relationship between the slope stability and influencing factors [21], which needs to be resolved. The use of multiple machine learning algorithms has always been accepted as a better solution, which can improve the nonlinear estimation ability of the model. Machine learning techniques have been widely used in theoretical and empirical approaches to geotechnical engineering and can achieve better performance than single weak learners, particularly when solving certain complex classification problems [22]. Machine learning methods such as random forest (RF) [23–25], support vector machine (SVM) [26–28], and extreme gradient boosting (XGBoost) [29,30] have been widely used to predict slope stability and landslide susceptibility in many studies. For example, Lee et al. [26] employed SVM for landslide susceptibility mapping in Gangwon Province, South Korea. Xu et al. [30] developed an ensemble learning approach based on a stacking strategy (XGBoost) to explore the feasibility of factor of safety prediction using dynamic multi-source monitoring data of slopes and landslides. Furthermore, previous studies have indicated that SVM has been effectively and widely applied as a reliable solution to many classification problems [31]. Previous studies have also indicated that bagging and boosting algorithms (RF and XGBoost) exhibit excellent performance in slope stability prediction and landslide susceptibility modeling [29,30,32]. Moreover, these machine learning algorithms have many adjustable parameters that are very important for proposing optimal models.

There has been no research on the assessment of slope stability using machine learning-based methods in the CACZ boreal forest ecosystem. Therefore, this study aims to provide a method for assessing the physical stability of forest restoration projects in the CACZ. Based on field survey data and machine learning, we developed four classification models: random forest (RF), support vector machine (SVM), extreme gradient boosting (XGBoost), and logistic regression (LR). Then, the effectiveness of LR was validated, the performance of the classification models was compared, and the feature importance of the 14 influencing factors was explored.

## 2. Materials and Methods

### 2.1. Investigated Data

The study area was located in the northern area of the Civilian Control Line, South Korea. The Korean Forest Service conducted a basic survey and data collection on the restoration area from 2009 to 2019. This area is under the jurisdiction of three metropolitan cities and provinces (Incheon Metropolitan City, Gyeonggi-do, and Gangwon-do). Two local forest offices (the Northern and Eastern Regional Offices of the Korean Forest Service) administer the national forests of these areas. There are also 12 divisions spread across the area because it is a military zone, and the study area is adjacent to tactical military roads in the CACZ. The area is approximately 87,863 ha (as of April 2020).

#### 2.1.1. Slope Conditions

For the study area, the altitude of the specific geographical location is accurately reflected by the elevation [29]. From Figure 2a, it can be seen that the elevation of the study area is mostly between 344 m and 790 m, and the average is 640 m.

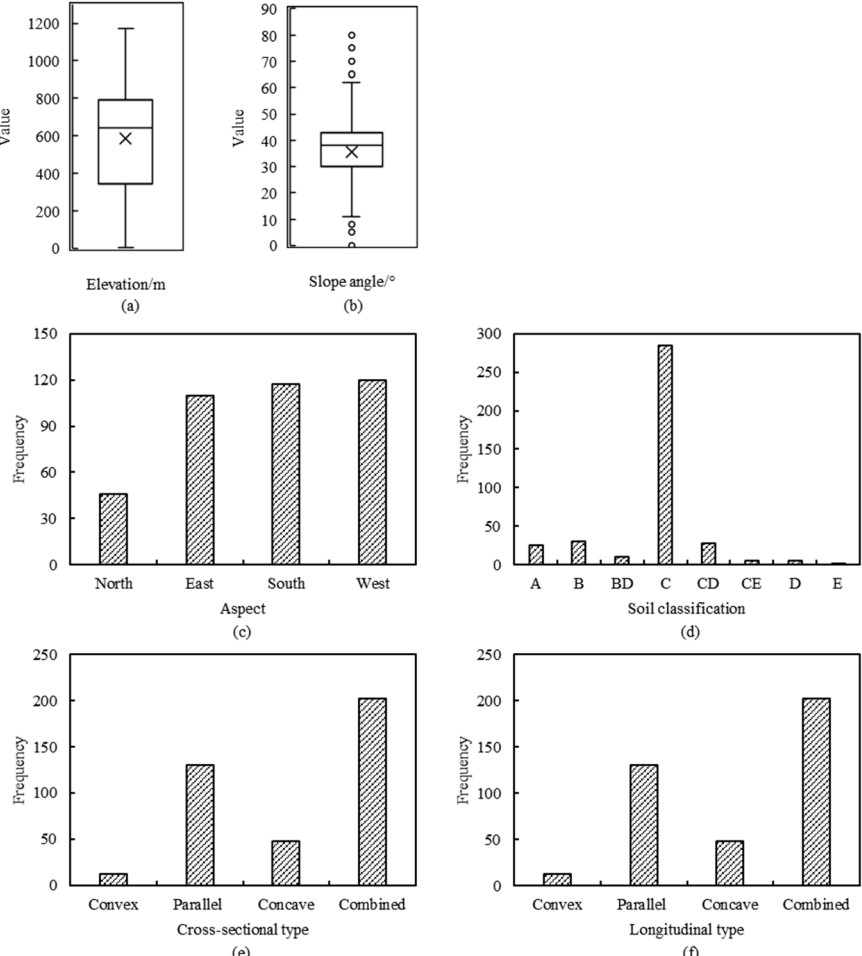

**Figure 2.** Distributions of slope condition features. (**a**) Elevation; (**b**) slope angle; (**c**) aspect; (**d**) soil classification; (**e**) cross-sectional type; and (**f**) longitudinal type. Soil classification: clay loam (A), loamy (B), loamy and weathered rock (BD), sandy loam (C), sandy and weathered rock (CD), sandy loam and soft rock (CE), weathered rock (D), and soft rock (E). The x in the box plot represents the mean of data; m, meter; °, degree.

The slope angle directly affects the amount of geomaterial deposited on a slope, and further affects the slope stability [29]. From Figure 2b, it can be seen that the slope angle of the study area is mostly between 30° and 43°, and the average is 38°.

As shown in Figure 2c, the slope aspect was divided into four categories: north-, east-, south-, and west-facing slopes; the south- and west-facing slopes accounted for the largest portion.

Soil can be classified into eight primary types based on its texture. As shown in Figure 2d, these types include clay loam (A), loamy (B), loamy and weathered rock (BD), sandy loam (C), sandy and weathered rock (CD), sandy loam and soft rock (CE), weathered rock (D), and soft rock (E). Sandy loam constituted the largest proportion (approximately 72%), followed by loamy (8%), sandy and weathered rocks (7%), and clay loam (6%).

Cross-sectional and longitudinal slope types can be classified as follows: convex, parallel, concave, and combined. Figure 2e,f plots the distributions of these four types. This shows that for both cross-sectional and longitudinal slope types, the combined type accounts for the highest proportion.

### 2.1.2. Vegetation Conditions

Vegetation plays an important role in slope stability and has a mechanical influence on the stability of the slope; that is, vegetation stabilizes the slope through mechanical reinforcement of the soil by the root system [33]. Vegetation works include seeding, sodding, slope mulching, tree planting, and combinations thereof. We classified the samples into two categories based on the absence or presence of vegetation: no and yes. As shown in Figure 3a, plots without vegetation works accounted for 34%, and more than half of the plots had vegetation works.

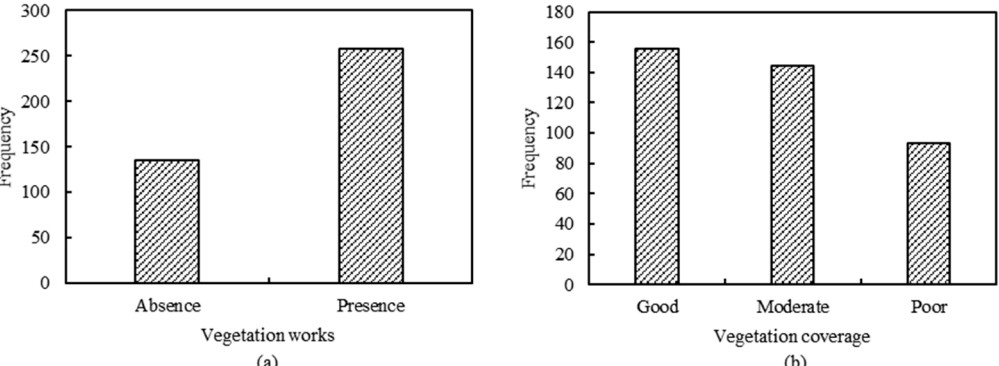

**Figure 3.** Distributions of vegetation condition features. (**a**) Vegetation works and (**b**) vegetation coverage.

Vegetation coverage is a critical indicator of vegetation growth and is divided into three categories according to vegetation coverage: good (>70%), moderate (40%–70%), and poor (less than 40%). Figure 3b shows the distribution of these three categories; as shown, the plots with good vegetation coverage were the most common, accounting for approximately 40%, followed by moderate (36%) and poor (24%) conditions.

### 2.1.3. Structure Conditions

Slope structures are categorized into three types according to their functions: slope stabilization works for supporting the slope, slope protection works for slope protection and vegetation, and surface water drainage works for surface water removal. The areas are categorized into two categories, presence or absence, based on the presence or absence of these works.

Figure 4a shows that the width of the structures is mostly between 20 m and 60 m, and the average is 35 m. The length (Figure 4b) of the structures is mostly between 10 and 31 m, with an average of 15 m.

Regarding slope stabilization (e.g., boulder stacking, retaining walls) (Figure 4c), 78% of the plots lacked slope stabilization works, whereas 22% of the plots included slope stabilization works. For slope protection works (e.g., mulching, stone masonry, and vegetation sack stacking) (Figure 4d), 20% of the plots lacked slope protection works, whereas 80% of the plots had these works constructed. For surface water drainage works (e.g., stone, sod, concrete channels, rill control, and diversion drains at the top of the slope) (Figure 4e), 80% of the plots lacked surface water drainage works, whereas 20% of the plots had these works constructed.

External conditions are categorized into three classes based on the structure's conditions, function, and slope conditions: good, moderate, and poor. Good conditions mean that there are no problems in the structures, such as cracks and broken areas, the disaster prevention function and safety function is in good condition, and the rills and drainage channels are slightly developed. Moderate conditions mean that there are some minor problems for the structures, such as cracks and broken areas, and there is no problem in function or safety, but the existing problems may continue to expand the risk, and the rills and drainage channels are developed on a small scale. Poor conditions mean that there

are many problems, such as cracks and broken areas in the structure or structural defects due to soil loss from the foundation of a structure, and the rills and drainage channels are developed on a large scale. Figure 4f plots the distribution of these three classes. It is shown that plots with good structure conditions are most common, accounting for about 46%, followed by moderate conditions (38%) and poor conditions (16%).

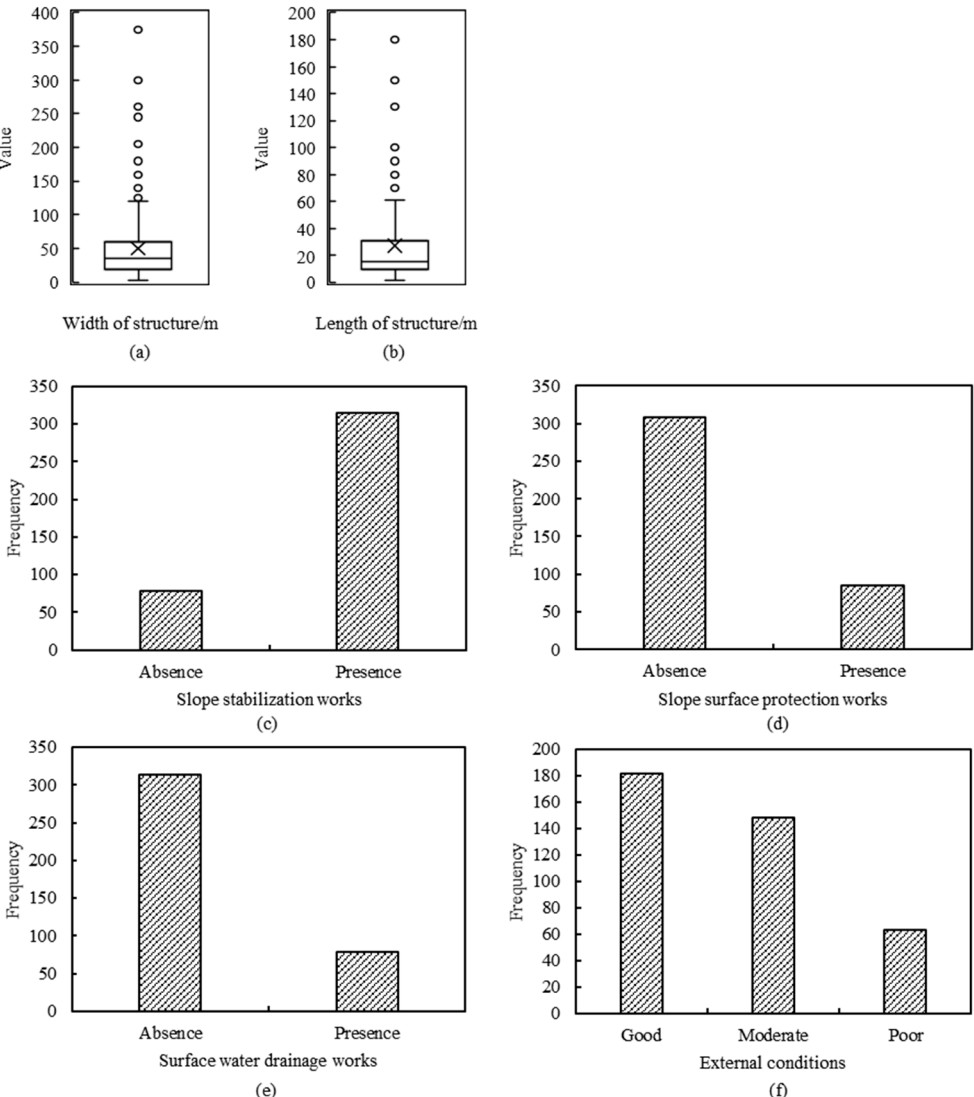

**Figure 4.** Distributions of structure condition features. (**a**) Width of the structure; (**b**) length of the structure; (**c**) slope stabilization works; (**d**) slope surface protection works; (**e**) surface water drainage works; and (**f**) external conditions. The x in the box plot represents the mean of data; m, meter.

### 2.2. Data Processing

To assess the slope stability of the mountainous restoration sites, it was necessary to select the most influential factors. For this purpose, 14 features were selected as input to establish the models. Descriptions of each selected feature are presented in Table 1.

Among the influencing factors selected in this study, there are four continuous variables (elevation, slope angle, width of structure, and length of structure) [16,29], and ten categorical variables (soil classification, cross-sectional, etc.). For the continuous features, they are not further coded as the established model can distinguish their magnitude. For the categorical features, since the model is not distinguishable from the text-descriptive features, each categorical feature will be numbered by the label encoding method [29]. In addition, in this study, data scaling was achieved by standardization.

The database consisted of 393 sample data points of slopes adjacent to military operation roads in the CACZ. The Korean Forest Service conducted a basic survey and collected data from the restoration area from 2009 to 2019. In addition, the survey plots in the area were classified into two grades for analysis: stable (228 cases) and unstable (165 cases). The two categories of slope stability cases were relatively balanced (stable 58%, unstable 42%) and were randomly divided into two groups of data with a ratio of 8:2 (Figure 5), of which 321 were grouped into the training set and 72 were grouped into the test set (Table 2).

**Table 1.** Description of selected features.

| Classification | Feature | Type | Description |
|---|---|---|---|
| Slope conditions | Elevation | Number | Meter (m) |
| | Slope angle | Number | Degree (°) |
| | Aspect | 4 categories | North, east, south, and west |
| | Soil classification | 8 categories | Clay loam (A), loamy (B), loamy and weathered rock (BD), sandy loam (C), sandy and weathered rock (CD), sandy loam and soft rock (CE), weathered rock (D), and soft rock (E) |
| | Cross-sectional type | 4 categories | Convex, parallel, concave, and combined |
| | Longitudinal type | 4 categories | Convex, parallel, concave, and combined |
| Vegetation conditions | Vegetation works | 2 categories | Presence and absence |
| | Vegetation coverage | 3 categories | Good, moderate, and poor |
| Structure conditions | Width of the structure | Number | Meter (m) |
| | Length of the structure | Number | Meter (m) |
| | Slope stabilization works | 2 categories | Presence and absence |
| | Slope surface protection works | 2 categories | Presence and absence |
| | Surface water drainage works | 2 categories | Presence and absence |
| | External condition | 3 categories | Good, moderate, and poor |

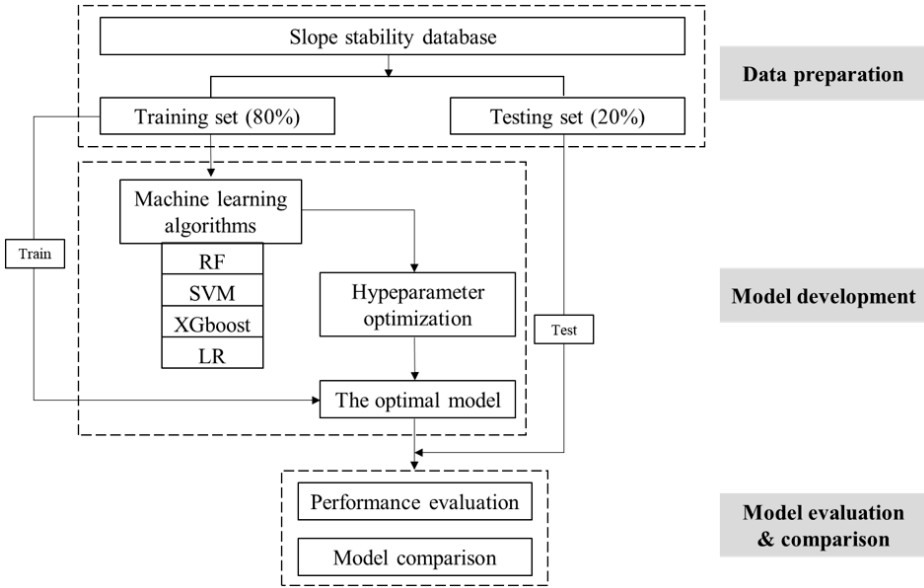

**Figure 5.** Flowchart of slope stability analysis using supervised learning methods.

**Table 2.** The slope stability data partition.

| Grade | Training Data | Test Data | Total |
|---|---|---|---|
| Stable | 188 (0.586) | 40 (0.556) | 228 (0.580) |
| Unstable | 133 (0.414) | 32 (0.444) | 165 (0.420) |
| Sum | 321 | 72 | 393 |

*2.3. Methodology*

2.3.1. Discrimination Methods

Supervised learning methods were used to simulate and predict slope stability, and four commonly used classification models were selected: random forest (RF), support vector machine (SVM), extreme gradient boosting (XGBoost), and logistic regression (LR). These were chosen for their advantages of mature theory and high efficiency [34].

1. Random forest (RF)

RF is an ensemble machine learning technique developed by Breiman [35] that comprises decision trees using bagging methods. As one of the most widely used classifier methods, RF has been successfully used for regression, classification, and feature selection, and represents an ensemble of individually trained binary decision trees [36]. Random forest is a tree-based machine learning algorithm that leverages the power of multiple decision trees to make decisions. During model building, RF creates multiple decision trees by randomly subsetting a predefined number of variables. The final prediction result is determined by a majority vote of all trees [37].

2. Support vector machine (SVM)

The SVM, introduced by Boser, Guyon, and Vapnik [38], is a widely used classification approach. As a supervised learning method, one of the key features of SVM is the ability to use different kernel functions to model nonlinear relationships between the input variables and the output variable. Here the default kernel Gaussian radial basis function is used as the kernel function. SVM has been widely utilized in different classification, pattern recognition, and regression problems because of its effectiveness in working with linearly non-separable and high-dimensional datasets [39,40]. It aims to identify a decision boundary with the largest possible margin that can still separate different classes [15].

3. Extreme gradient boosting (XGBoost)

XGBoost is another ensemble learning technique (specifically, a boosting technique) proposed by Friedman [41], which creates a prediction model utilizing weak prediction techniques such as decision trees [16]. XGBoost is an improvement of the gradient boosting algorithm. Newton's method is used to solve for the extreme value of the loss function, and the Taylor loss function is expanded to the second order. In addition, a regularization term is added to the loss function. The objective function during training consists of two parts: the gradient boosting algorithm loss and the regularization term [42].

4. Logistic regression (LR)

LR refers to a type of generalized linear model used to describe data and estimate the probability of a binary response based on one or more nominal, ordinal, interval, or ratio-level independent variables [43]. A binary logistic regression model has a dependent variable with two values: 0 and 1.

2.3.2. Parameter Optimization

The complexity of a model is determined by its hyperparameters, which are the key elements of the model. Determining the best combination of these parameters is critical for optimizing the model [44]. Of the four aforementioned model types, the hyperparameters for RF, SVM, and XGBoost have been optimized to improve the proposed models. To achieve high performance for these machine learning models, we need to tune their hyperparameters. Hyperparameters are parameters of a model that are not trained from data and are used to configure the model (e.g., number and depth of decision trees) [45]. The hyperparameters of the different models that we mainly considered are as follows:

Random forest (RF) model: The hyperparameters of the RF algorithm are mtry and ntrees; mtry defines the number of variables randomly sampled as candidates at each split, and ntrees defines the number of trees to grow. The optimal hyperparameters for mtry and ntree were 4 and 200, respectively.

SVM model: The hyperparameters of the SVM algorithm are the penalty coefficient *C* and gamma ($\gamma$). The hyperparameter *C* reflects the tolerance of the SVM model to errors, and the hyperparameter $\gamma$ defines how far the influence of a single training example reaches. Based on the 10-fold CV method, the range of values tuned for *C* and $\gamma$ are $2^{-2}$:$2^{10}$, and $2^{-10}$:$2^5$, respectively, and the optimal values for *C* and $\gamma$ are 4 and 0.125, respectively. In addition, the default kernel Gaussian radial basis function was used as the kernel function in this study.

XGBoost model: The hyperparameters of the XGBoost algorithm include the learning rate (Eta), minimum loss during splitting (gamma), maximum tree depth (max depth), minimum sum of weights (min child weight), number of rounds (nrounds), column subsampling parameters (colsample bytree), and ratio considered for model training (subsample). Based on the highest accuracy values from all combinations, the XGBoost model was optimized by fixing values of 0.1, 0, 10, 1, 100, 0.5, and 1 for Eta, gamma, max depth, min child weight, nrounds, colsample bytree, and subsample, respectively.

### 2.3.3. Performance Metrics

The confusion matrix is a basic tool for evaluating the performance of supervised ML algorithms. In this study, the confusion matrix is a 2 × 2 matrix (Table 3); according to this matrix, certain metric values (e.g., accuracy, recall rate, precision, and F1-score) can be estimated [46].

**Table 3.** Confusion matrix.

| Predicted Class | Actual Class | |
| --- | --- | --- |
| | Positive | Negative |
| Positive | True positive (TP) | False positive (FP) |
| Negative | False negative (FN) | True negative (TN) |

In addition, the receiver operating characteristic (ROC) curve was used, which plots the true sensitivity or positive rate against 1-specificity or false positive rate and can be used to evaluate the performance of the classification performance [47]. Moreover, the area under the ROC curve (AUC) is also commonly used as a metric to evaluate the prediction accuracy of classifiers. The value of the AUC ranges from 0.5 to 1, and the relationship between the AUC and discrimination accuracy can be interpreted using five ratings [47]: no discrimination (0.5–0.6), poor discrimination (0.6–0.7), fair discrimination (0.7–0.8), good discrimination (0.8–0.9), and excellent discrimination (0.9–1.0). In addition, in order to measure the degree of agreement between the ratings assigned by the two groups, the interrater reliability was assessed through Cohen's kappa, which is a measure of interrater reliability used to measure agreement between two coders [48]. Kappa values range from −1 to 1, with 1 indicating complete agreement and 0 meaning no agreement or independence. According to Landis and Koch [49], these values can be interpreted as follows: kappa values of 0.61 to 0.8 indicate substantial agreement, and values of 0.8 to 1.0 suggest almost perfect agreement.

## 3. Results

### 3.1. Predictive Performance of Different Machine Learning Models

Figure 6 shows the confusion matrix of the RF model. For the training data, the accuracy, recall rate, and precision of the two conditions were all 1.000. All stable and unstable conditions were correctly identified. For the testing data, the accuracy was 0.875, and the recall rates for the stable and unstable conditions were 0.850 and 0.906, respectively. Six stable conditions were mistakenly identified as unstable, and three unstable conditions were incorrectly predicted as stable. As shown, the prediction results of the RF model on the training data are perfect.

|  | | Actual class | | |
|---|---|---|---|---|
|  |  | Stable | Unstable | Precision |
| Predicted condition | Stable | 188 | 0 | 1.000 |
|  | Unstable | 0 | 133 | 1.000 |
|  | Recall | 1.000 | 1.000 | 1.000 |

(a)

|  | | Actual class | | |
|---|---|---|---|---|
|  |  | Stable | Unstable | Precision |
| Predicted condition | Stable | 34 | 3 | 0.900 |
|  | Unstable | 6 | 29 | 0.923 |
|  | Recall | 0.850 | 0.906 | 0.875 |

(b)

**Figure 6.** Confusion matrix of RF algorithm prediction values. (**a**) Training data and (**b**) testing data.

Figure 7 shows the confusion matrix of the SVM model. For the training data, the accuracy and recall rates of the two conditions (stable and unstable) were 0.928, 0.956, and 0.893, respectively. For the testing data, the accuracy was 0.847, and the recall rates for the two conditions were 0.837 and 0.862, respectively. Seven stable conditions were mistakenly identified as unstable, and four unstable conditions were incorrectly predicted as stable. Thus, the performance of the SVM was similar to that of the RF model.

|  | | Actual condition | | |
|---|---|---|---|---|
|  |  | Stable | Unstable | Precision |
| Predicted condition | Stable | 173 | 15 | 0.920 |
|  | Unstable | 8 | 125 | 0.940 |
|  | Recall | 0.956 | 0.893 | 0.928 |

(a)

|  | | Actual condition | | |
|---|---|---|---|---|
|  |  | Stable | Unstable | Precision |
| Predicted condition | Stable | 36 | 4 | 0.900 |
|  | Unstable | 7 | 25 | 0.781 |
|  | Recall | 0.837 | 0.862 | 0.847 |

(b)

**Figure 7.** Confusion matrix of SVM algorithm prediction values. (**a**) Training data and (**b**) testing data.

Figure 8 shows the confusion matrix of the XGBoost model. For the training data, the accuracy, recall rate, and precision of the two conditions were all 1.000. All stable and unstable conditions were correctly identified, as in the RF model. For the testing data, the accuracy was 0.903, and the recall rates for the two conditions were 0.902 and 0903, respectively. Four stable conditions were mistakenly identified as unstable, and three unstable conditions were incorrectly predicted as stable. It can be observed that the performance of the XGBoost model was better than that of the RF and SVM models.

Figure 9 summarizes the confusion matrix obtained from the LR model. For the training data, the accuracy and recall rates of the two conditions (stable and unstable) were 0.903, 0.920, and 0.881, respectively. For the testing data, the accuracy was 0.847 and the recall rates for the two conditions were 0.822 and 0.889, respectively. Eight stable conditions were mistakenly identified as unstable, and three unstable conditions were incorrectly predicted as stable. In general, the performance of the LR model was similar to that of the SVM model, and the XGBoost model outperformed the other three models.

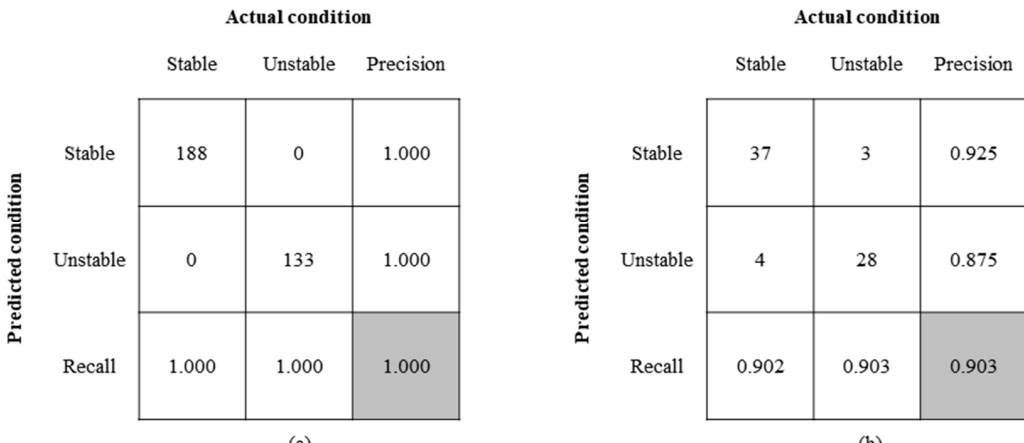

**Figure 8.** Confusion matrix of XGBoost algorithm prediction values. (**a**) Training data and (**b**) testing data.

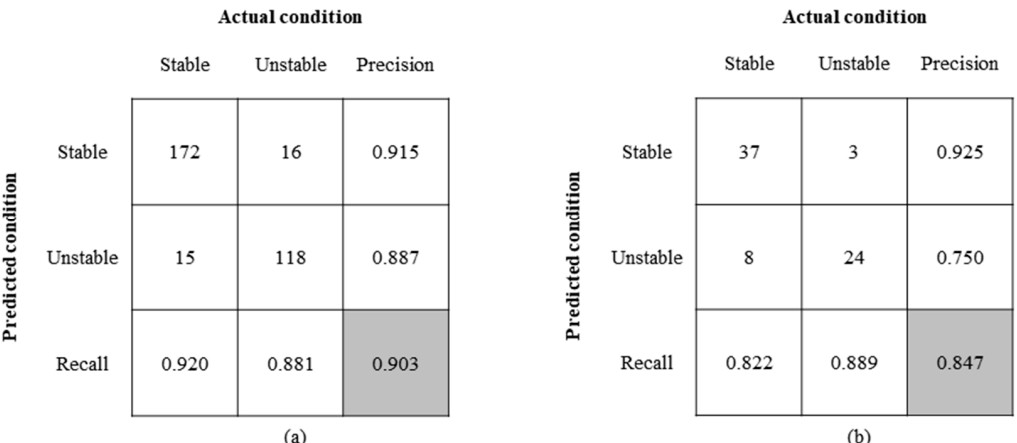

**Figure 9.** Confusion matrix of LR algorithm prediction values. (**a**) Training data and (**b**) testing data.

According to the accuracies of these four models, RF and XGBoost perform better than SVM and LR when predicting the stability of the slopes adjacent to military tactical s roads in the CACZ. In machine learning, overfitting is a common phenomenon in which a model fits the training data too well and, as a result, is unable to accurately predict on test data. In other words, overfitting may occur if there is a significant difference in the accuracies between the training data and test data [29]. In addition, the differences between the prediction accuracies of the training and test data are 0.125 (RF), 0.081 (SVM), 0.097 (XGBoost), and 0.056 (LR), which are less than 0.15. According to Hou et al. [34], there is still no exact reference regarding to what extent differences between the training and test data indicate overfitting; in this study, the difference was relatively small.

The performance of the four models for the testing dataset is shown in Table 4. Between these models, according to the F1-score of the testing dataset, XGBoost (F1-score = 0.914) had the highest model accuracy among the models, followed by RF (F1-score = 0.883), LR (F1-score = 0.871), and SVM (F1-score = 0.867). As shown, the kappa values of the models varied from 0.688 to 0.803, and XGBoost had the highest kappa value (0.803), indicating an almost perfect agreement. Furthermore, XGBoost also had the highest recall rate (0.902). XGBoost and LR had the same precision value (0.925).

**Table 4.** Evaluation metrics of the four ML algorithms.

| ML Algorithm | F1-Score | Kappa Value | Recall Rate | Precision |
|:---:|:---:|:---:|:---:|:---:|
| RF | 0.883 | 0.749 | 0.850 | 0.919 |
| SVM | 0.867 | 0.688 | 0.837 | 0.900 |
| XGBoost | 0.914 | 0.803 | 0.902 | 0.925 |
| LR | 0.871 | 0.686 | 0.822 | 0.925 |

The AUC scores of the four ML models are shown in Figure 10. It can be seen that the AUC value of the testing dataset for XGBoost is 0.900, which could be considered to indicate excellent classification performance (AUC > 0.9), followed by RF (AUC = 0.878), SVM (AUC = 0.841), and LR (AUC = 0.838).

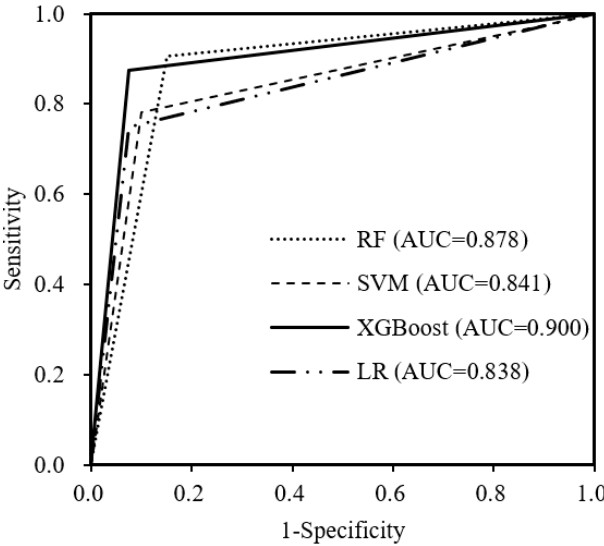

**Figure 10.** ROC curves and corresponding AUC values for the four models.

In general, among the four predictive models, XGBoost performed better than RF, SVM, and LR in assessing the stability of the mountainous restoration sites assessment in the northern area of the Civilian Control Line.

*3.2. Feature Importance Analysis*

Feature importance analysis was performed to investigate the role of each feature. Feature importance is an important reference for feature selection and model interpretability. A trained XGBoost-based model can automatically calculate the feature importance, which can be obtained through the interface feature importance criterion, that is, the gain criterion. The gain is calculated by taking each feature's contribution to each tree in the model; the higher the relative importance of the feature, the more the feature contributes to the model [50]. The importance of the 14 features in the XGBoost model is ranked in Figure 11. The external condition and vegetation coverage were the most important variables, with importance values of 38.5% and 15.2%, respectively, followed by elevation (11.9%), slope angle (11.2%), length (6.1%), and width (6.0%). This result provides an important reference for exploring the stability evolution of landslides and provides multisource monitoring data for slope restoration.

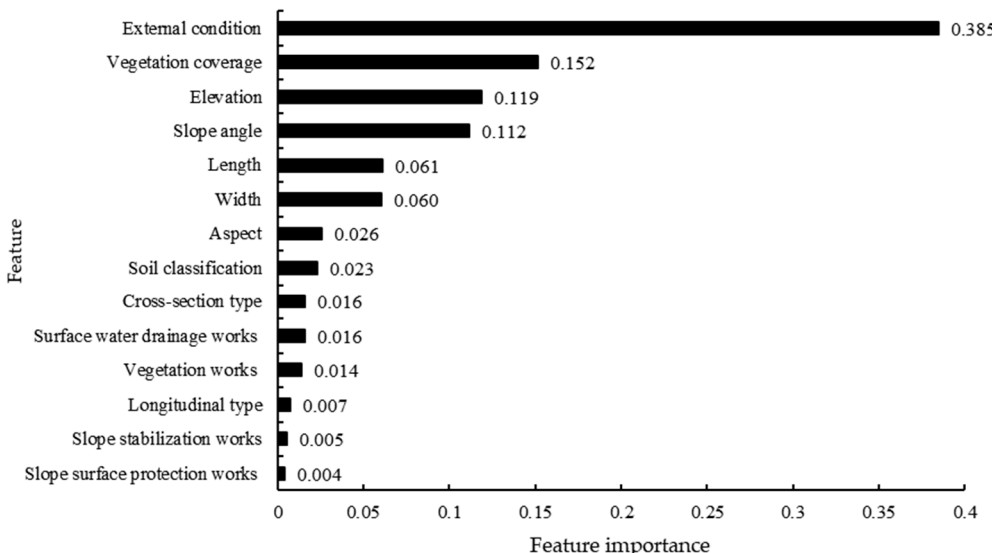

**Figure 11.** Feature importance of influencing factors.

## 4. Discussion

In this study, supervised RF, SVM, XGBoost, and LR methods were developed to predict slope stability. RF and XGBoost were shown to be superior to the conventional methods (i.e., SVM and LR) [29]. This conclusion is basically the same as in the literature [29]. This may be because the model inherits the powerful learning ability of multiple individual learners so that the ensemble learning models can approximate the implicit high-dimensional relationship between the various influencing factors and slope stability in this study [29]. XGBoost performed the best among these four methods in the stability assessment of slopes adjacent to military tactical roads within the CACZ, South Korea. Compared with random forest, XGBoost, as a tree-based model, includes more tuneable hyperparameters, and in boosting, boosted trees are grown sequentially. Specifically, each of the trees is grown using information from previously grown trees, unlike bagging, where multiple copies of original training data are created and fitted to separate decision trees [51]. This may explain why XGBoost generally performs better than random forest. In contrast to SVM, XGBoost attempts different paths when it encounters a missing value on each node and learns which path to use to handle missing values [52]. In addition, SVM does not perform well with missing data; therefore, it is better to impute the missing values before running SVM. Furthermore, as for LR, if the relationship between the features and slope stability is well approximated by a linear model, linear regression may be a strong candidate.

Among the 14 influencing factors, the external condition was determined to be the most important based on the feature importance analysis in the XGBoost-based method. This was followed by vegetation coverage, elevation, and slope. These results are different from those of general forest slopes; the structure of the slopes in this area (CACZ) is limited owing to military security restrictions, such as construction materials, construction methods (stone stacking), etc. Thus, slope management is relatively difficult, and can only be completed on a small scale or with simplified materials and construction methods [53]. In addition, enhancing the stability of slopes through comprehensive reforestation is difficult because of the need to ensure visibility in military areas. Therefore, in this area, the external condition and vegetation coverage are the most important factors affecting slope stability. The importance of altitude and slope were third and fourth, respectively, because military installations (e.g., military roads) are concentrated at relatively high altitudes in this area [54] for the convenience of military operations. Therefore, elevation has a great impact on slope stability. The slope angle, similar to that of a general forest area, is a very important factor affecting the stability of the slope, and the variable importance analysis showed a similar result in this area [55]. In addition, other influencing factors, such as aspect, soil classification, cross-sectional type, longitudinal type, vegetation works, slope stabilization

works, slope surface protection works, and surface water drainage works, affect the stability of slopes in this area through various complex interactions. Different influencing factor rankings may be obtained when different datasets and classification models are analyzed in different study areas [56]. In addition, previous studies have paid more attention to the mechanical parameters and geometric variables (e.g., slope height, slope angle, cohesion, the pore water ratio, internal friction, and the unit weight) [16,53,57]; for example, Yang et al. [57] showed that cohesion was the most sensitive factor affecting slope stability, followed by slope height, rock bulk density, and slope angle. Wang et al. [58] found that the most important factor influencing slope stability is slope height, followed by cohesion, internal friction angle, and slope angle. However, it is well recognized that slope stability is affected by many factors such as mechanical parameters, geometric variables, topographic features, and geological conditions [29]. Therefore, in future research, for improving the assessment of slope stability, it is necessary to consider these comprehensive factors.

The performance of LR (accuracy = 0.847; AUC = 0.838) was good, and from the results of the feature importance analysis, the two most important factors concluded through LR analysis by the Korean Forest Service [13] were the same as those in the XGBoost-based analysis. Thus, the effectiveness of LR in predicting slope stability in this area was verified to be acceptable.

This study did not need to consider the influence of sample imbalance on the prediction effect. According to the above analysis, there were no significant differences in the number of samples with the two slope stability conditions, as shown in Table 2, and the ratio of the training data to test data was approximately 8:2 (321/72). In the training dataset, the ratio of stable and unstable conditions was approximately 3:2(188/133), and in the test dataset, these two conditions were approximately 5:4 (40/32), which may lead to better fitting and prediction performance for samples. Thus, the overfitting effect in this study can be ignored. However, as for the k-fold cross-validation technique used during the tuning of the hyperparameters, the k value defined is 10; this may be a little large for our small sample dataset. If the k value is too large, this will lead to less variance across the training set and limit the model currency difference across the iterations [58]. Therefore, for a binary task, this seems to work by ignoring sample imbalance. However, this would not be true for multiclassification tasks. Commonly, the best CV for training and expecting to obtain a good generalization model is defined by using the stratified k-fold.

Our study has some limitations. First, in addition to the 14 influencing factors considered, the mechanical parameters (e.g., cohesion) [59–61] may also affect slope stability. Because the information on slope height and cohesion was not recorded, this may lead to certain limitations in determining factors that affect slope stability. Thus, in the future, adding these factors to the assessment may lead to more effective conclusions and provide a better database for restoration planning. Second, our small sample size of approximately 400 cases was one of the most crucial problems in the analysis. Machine learning techniques generally benefit from a large amount of data, which increases their performance [62]. However, we benefited from the detailed and high-quality data of the CACZ, which has many limitations in conducting surveys owing to military security restrictions.

## 5. Conclusions

In this study, based on 393 slope stability cases, an empirical comparison of four supervised learning methods (RF, SVM, XGBoost, and LR) was applied to assess slope stability in the CACZ, South Korea. Furthermore, 14 influencing factors (slope, vegetation, structure conditions, etc.) were considered to explore the feature importance of the XGBoost model, which exhibited the best performance. The conclusions are as follows.

(1) Among the four algorithms, according to the performance metrics, RF and XGBoost performed better than SVM and LR in the predictive analysis of slope stability. The effectiveness of LR was validated with an accuracy, kappa value, and AUC of 0.847, 0.686, and 0.838, respectively. Furthermore, the accuracy, kappa value, and AUC of XGBoost on

the test data were 0.903, 0.803, and 0.900, respectively; thus, XGBoost could be considered the best model for prediction.

(2) Among the 14 influencing factors, according to the feature importance analysis results obtained from XGBoost, the external condition and vegetation coverage were the most important variables, similar to the results of the analysis by the South Korean Forest Service using the LR method.

The proposed supervised learning-based method may also be applied to other landslide-prone areas. The analysis of the importance of influencing factors provides a database for restoration plans, which is necessary to understand the dynamics of damaged land and formulate a systematic management plan to prevent the expansion of damage.

**Author Contributions:** Conceptualization, S.K., L.P. and J.I.S.; methodology, S.K.; software, L.P.; validation, S.K., L.P. and J.I.S.; formal analysis, L.P.; investigation, Y.K., S.I.L. and K.Y.; resources, H.K., K.L. and J.I.S.; writing—original draft preparation, S.K. and L.P.; writing—review and editing, J.I.S.; visualization, H.K., K.L. and K.Y.; supervision, J.I.S.; project administration, Y.K.; funding acquisition, S.I.L. All authors have read and agreed to the published version of the manuscript.

**Funding:** This work was supported by a research grant from Kongju National University in 2022.

**Data Availability Statement:** Not applicable.

**Acknowledgments:** The authors would like to thank the staff of the CCZ Forest Land Management Office, Korea Forest Conservation Association, for all of their valuable comments and support.

**Conflicts of Interest:** The authors declare no conflict of interest.

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
