# Peer review of "Empirical Comparison of Supervised Learning Methods for Assessing the Stability of Slopes Adjacent to Military Operation Roads"

_forests, doi:10.3390/f14061237_

Round 1
Reviewer 1 Report
Dear Authors
The whole introduction was assigned to the study area, approximately. I left my comments in the text.

Reviewer 2 Report
Dear Editors,
I have read the manuscript titled “Empirical Comparison of Four Supervised Learning Methods for Assessing Stability of Slopes Adjacent to Military Operation Roads within South Korea’s Civilian Access Control Zone”. This is an interesting case study especially when using machine learning methodologies for solving land stability. The work can be a field-guide application for many other problems related to the environment and agriculture. I also understand the small set of datasets mainly due to military security restrictions. However, I have some comments to improve the manuscript which is already good enough based on its contents for publication after addressing my comments below. I also attached some basic comments inside the MS.
1) One issue I met is that sentences are sometimes long (e.g., Line 75-77, etc). I suggest the authors polish a bit more the MS by dividing as possible some long sentences into different short sentences to ease reading.
2) Line 92 – 98. I strongly agree with that section, but I would like to know why do you not use an ensemble stacking strategy to aggregate the performance of LR, RF, SVM, and XGBoot at final? I’m pretty sure, it will give you a better score greater than 90 % ( threshold of XGBoost) (e.g., Zhou 2007, Yin et al. 2021, Zounemat-kermani et al. 2021).
Indeed, Although RF and XGBoost are ensemble learning methods (bagging or pasting/boosting), the stacking approach at the final is a good tip for algorithms to correct individual mistakes each other to reach a final optimal score. When I checked the score and the selected algorithms for this workflow, they obey the three criteria ( wisdom of the crowd(Surowiecki 2005) ) for using the ensemble stacking strategy: (i) the diversity (SVM, LR even though both are parametric models, they operate differently), RF and XGBoost are non-parametric based on Trees but they under the hood scenario are different. (ii) The level of score: All scores are greater than 50 % and (iii) the popularity: Four paradigms (LR, SVM, RF, and XGBoost ) are acceptable for stacking
Can you convince me why did not use the Ensemble stacking strategy at the end to push a little bit your score? Or do you think that 90% of XGBoost is enough score? All scientists intend to get 99.99%.
3) Line 136-137 “Sandy loam constituted the largest proportion (approximately 72%), 136 followed by loamy (8%), sandy and weathered rocks (7%), and clay loam (6%).”. When normalizing the data into a percentage, the sum of labels must be equal to 100%. Here we got 93% for C, B, D, and A.
You may specify the ratio of BD, CD, and CE even if their ratios are not significant. This is useful to know whether there is also missing data (of course not applicable in cases related to soil classification since the soil whatever it is has a classification type).
As a suggestion for future work, when you faced a situation like the soil types classification AND dealing with a small sample of datasets like this workflow. Labels like BD, CD, and CE (=7%) can be merged either in B, C, or E rather than defining them separately to keep a relative balance proportion of datasets. For instance, in BD, you can setup a threshold of representativity equals to 50 % and if the composition of B > 50%, BD could be merged to B and vice versa). This is tip to have a balance label in soil classification and will facilitate the training phase thereby yielding a better bias and variance for a better generalization model.
As you know, the objective when using machine learning for this kind of topic is not to get a good score BUT the capability of the model to generalize into the unseen data with that score approximatively got during the training phase.
4) About the features “Cross-sectional and longitudinal slope types”, why don’t combine both types into a unique feature since both shared the same labels (convex, Parallel, Concave, Combined)? Is there any big difference between them or do you want to figure out something special, for instance, an interpretation purpose?
5) Section 2.1.3.: Structure conditions”
The figure 4f shows external conditions, However the explanation is missing in the text. After the fig 4e (line 182), you may explicitly give a breve explanation to what people must understand by external conditions. This will help reader to understand the choice of the labels “good – moderate- and poor”.
6) Section 2.2. Data preprocessing. The dataset is composed of numerical features (elevation, slope angle) and categorical features (soil classification, cross sectional etc.). Indeed, for tree-based algorithms like RF and XGBoost, we don’t need much processing however for LR and SVM, you must explain how did you encode the nominal labels to numeric to compose a unique predictor X before feeding to the algorithms and what is the new size of your dataset. For instance, the use of one-hot-encoding strategy. You can have an example from these authors (Geron 2019, Wlodarczak 2019, Kouadio et al. 2022). It is a good place to explain whether is there any missing data and how did you impute them. Did you scale the data? if yes, explain the type of scaling did you use.
After including this part, change the section name “2.2. Data preprocessing” to “2.2. Data processing”
7) Line 204-218 and by extension in the whole manuscript, I realized that you use the SVM without “s” at the end. That mean you uses one kernel of SVMs. You may specify which kernel of SVMs did you use? (Radial basis function (RBF), sigmoid, linear, polynomial, etc.). If you used scikit-learn (Buitinck et al. 2013) , the default kernel is RBF. However, if you developed your own kernel, you may specify. This is an important to know how you optimized the hyperparameters to.
You can specify the SVM kernel in line 230 when giving a breve illustration of SVMs.
Line 230 “The SVM, introduced by Boser, Guyon, and Vapnik [34] “
For your information, I think SVMs were first introduced in 1970s and early 1980s by Vladimir Vapnik and became famous as a classifier method based on the structural risk minimization principle before the Proceedings of Boser and al. in 1992 during the fifth annual workshop as an approach to binary classification. So, I understand when it is sometimes confused in the literature with authors used verb “introduced”. You can keep it in mind this little story.
8) Line 240: “proposed by Chen and Guestrin”. I recommend to cite (Friedman 2001) as the first proposer of XGBoost before Chen T.; Guestrin C in 2016.
9) Line 253.” LR refers to a type. “ cite (Cox 1958) as the first implementor.
10) Line 261-263: Even LR has some hyperparameters that can be fine-tuned such as 'l1' or 'l2' as penalty and C as inverse of regularization strength. By default, l2 is used in scikit-learn. If you have not optimized the LR hyperparameters, please reformulate this sentence: “Of the four aforementioned model types, three (RF, SVM, and XGBoost) have hyperparameters “
11) Line 273-275. SVM has also an additional hyperparameter called slack variable(
which measures how much an instance is allowed to violate the margin. Since you fine-tuned gamma
that means the linear kernel is excluded so you need to specify which kernel did you used as I mentioned earlier.
12) ) Line 287 -293 : You can add the following references to support the literature about Precision recall metrics(Powers 2007) and I appreciate the nice flow chart in Figure 5.
13) Line 239, 330: I suggest to display the learning curves of the fourth models. You can use your own code our the one developed by these authors (Kouadio et al. 2023) to inspect multiple models at once.
In addition, the score of 100% is quite normal since the CV=10 is large for a small sample of dataset. Some fold score can easily give 100 % or low than this. Commonly the best CV for training and expecting to get a good generalization model is how the data can be more representative at the same time in the training and test sets using the stratified k-fold. For a binary task as indicated in table 2 (based on the ratio), this seems work by ignoring sample imbalance. However, this would not be true for multiclassification task. This as a good idea to mention it in the discussion section (Line 453-455)
14) Line 402-413. After addressing my comment 6), use XGBoost and RF applied to the transformed dataset to rebuild both feature importance schemes and confirm whether the external condition is still having the highest rate.
As a perspective for this work and after reading some of your references, I realized that the times series work using this kind of dataset is not done yet. I recommend IF there is an additional data that include seasonal variation observed during the day, month and years (like impact of the rain, wind direction, etc.), to try the prediction of the land stability during the seasonal variation (or moment of times). This can be another topic very interesting as a field-guide for the Korean Forest Service to prevent the risk of land subsidence which is also one aspect of slope stability.
References
Buitinck, L., Louppe, G., Blondel, M., Pedregosa, F., Mueller, A., Grisel, O., Niculae, V., et al. (2013) API design for machine learning software: experiences from the scikit-learn project. ECML PKDD Work. Lang. Data Min. Mach. Learn., pp. 108–122.
Cox, D.R. (1958) The regression analysis of binary sequences. J. R. Stat. Soc. Ser. B, 20, 215–232, Wiley Online Library.
Friedman, J.H. (2001) Greedy function approximation: a gradient boosting machine. Ann. Stat., 1189–1232, JSTOR.
Geron, A. (2019) Hands-on machine learning with Scikit-Learn, Keras, and TensorFlow: concepts, tools, and techniques to build intelligent systems. (N. Tache, N. Adams, R. Monaghan & R. Charles, Eds.)O’Reilly Media Inc, Sebastopol, CA., 1rst ed., O’Reilly Media, Inc., 1005 Gravenstein Highway North, Sebastopol, CA 95472. Retrieved from https://upload.houchangtech.com/pdf/Hands-on_Machine_Learning.pdf
Kouadio, K.L., Liu, J. & Liu, R. (2023) watex: machine learning research in water exploration. SoftwareX, 22, 101367, Elsevier B.V. doi:10.1016/j.softx.2023.101367
Kouadio, K.L., Loukou, N.K., Coulibaly, D., Mi, B., Kouamelan, S.K., Gnoleba, S.P.D., Zhang, H., et al. (2022) Groundwater Flow Rate Prediction from Geo‐Electrical Features using Support Vector Machines. Water Resour. Res., 1–33. doi:10.1029/2021wr031623
Powers, D.M.W. (2007) Evaluation: From Precision, Recall and F-Factor to ROC, Adelaide. Retrieved from https://www.scinapse.io/papers/46659105#fullText
Surowiecki, J. (2005) The wisdom of crowds, 6th ed., New York : Anchor Books, c 2005 (OCoLC)1085906407.
Wlodarczak, P. (2019) Machine Learning and its Applications. Mach. Learn. its Appl. doi:10.1201/9780429448782
Yin, J., Medellín-azuara, J., Escriva-bou, A. & Liu, Z. (2021) Science of the Total Environment Bayesian machine learning ensemble approach to quantify model uncertainty in predicting groundwater storage change. Sci. Total Environ. J., 769, 12. doi:10.1016/j.scitotenv.2020.144715
Zhou, Z. (2007) Ensemble Learning, Nanjing. Retrieved from file:///C:/Users/Administrator/Downloads/springerEBR09.pdf
Zounemat-kermani, M., Batelaan, O., Fadaee, M. & Hinkelmann, R. (2021) Ensemble machine learning paradigms in hydrology : A review. J. Hydrol., 15, 126266, Elsevier B.V. doi:10.1016/j.jhydrol.2021.126266
Minor editing of English language required
Reviewer 3 Report
The topic of this paper is interesting- landslide assessment in the Civilian Access Control Zones in the middle of Korea. On the other hand, the 4 supervised Learning models are common, and the result cannot give any suprise to readers. There are 4 parts should pay attention.
1. Figure 10, it is the first time to see an AUC with straight line.
2. As a landslide assessment paper, a map with the assessment result should be provided.
3. The external condition is the most important factor in the paper, but it lack a clear definition and explanation.
4. the influencing factors use soil classification, the author may think about the depth of soil or the depth of interface between bedrock and soil.
Round 2
Reviewer 1 Report
Dear Authors
Thank you for conducting this study. I have put my comments in the text.
